# Exogenous Application of Thiourea for Improving the Productivity and Nutritional Quality of Bread Wheat (*Triticum aestivum* L.)

**Ahmad Sher** [1], **Xiukang Wang** [2,*] , **Abdul Sattar** [1], **Muhammad Ijaz** [1], **Sami Ul-Allah** [1] , **Muhammad Nasrullah** [1], **Yamin Bibi** [3], **Abdul Manaf** [4], **Sajid Fiaz** [5] and **Abdul Qayyum** [6,*]

1 College of Agriculture, Bahauddin Zakariya University, Bahadur Sub-Campus Layyah, Layyah 31200, Pakistan; ahmad.sher@bzu.edu.pk (A.S.); abdulsattar04@bzu.edu.pk (A.S.); muhammad.ijaz@bzu.edu.pk (M.I.); samipbg@bzu.edu.pk (S.U.-A.); m.nasrullah8876@gmail.com (M.N.)
2 College of Life Sciences, Yan'an University, Yan'an 716000, China
3 Department of Botany, PMAS-Arid Agriculture University, Rawalpindi 46000, Pakistan; dryaminbibi@uaar.edu.pk
4 Department of Agronomy, PMAS-Arid Agriculture University, Rawalpindi 46000, Pakistan; drmunaf@uaar.edu.pk
5 Department of Plant Breeding & Genetics, The University of Haripur, Haripur 22620, Pakistan; sfiaz@uoh.edu.pk
6 Department of Agronomy, The University of Haripur, Haripur 22620, Pakistan
* Correspondence: wangxiukang@yau.edu.cn (X.W.); aqayyum@uoh.edu.pk (A.Q.)

**Abstract:** Because it is a staple food, sustainable production of wheat is crucial for global food security. Arid and semi-arid regions are worst affected by climate change, which has resulted in poor productivity of different crops, including wheat. To this end, this study aimed to investigate the effect of foliage-applied thiourea on the growth, yield, and nutritional-quality-related traits of bread wheat. The treatments consisted of thiourea levels (control, 500, and 1000 mg L$^{-1}$) factorally combined with two diverse wheat cultivars (Gandam-1 and Galaxy-2013) at different growth stages (tillering, booting, and heading) and was repeated over two years. The analysis of the data shows that thiourea treatments and the cultivars significantly ($p \leq 0.05$) affected the growth, nutritional quality traits, and morphological traits, and the interaction of the two factors was also significant. Improvement in productivity and nutritional quality was observed from the application of thiourea in both cultivars. Galaxy-2013 performed best at 1000 mg L$^{-1}$ thiourea application for both productivity- and nutritional-quality-related traits at the heading stage. In conclusion, exogenous application of thiourea improves the productivity and nutritional quality of wheat on sandy loam soils in semi-arid regions; however, for wider recommendations, more trials may be conducted across various agro-ecological regions.

**Keywords:** thiourea levels; seed yield; nutritional quality traits; wheat varieties

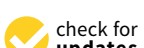

## 1. Introduction

When considering agricultural policies, the main focus is on wheat, as it is a very crucial crop and is the main crop of Pakistan. The contribution of wheat to GDP is 1.7%, and its contribution to value added in agriculture is 8.7%. During the years 2019–2020, the area under cultivation of wheat was 8825 thousand hectares. The average yield of wheat was 2827 kg ha$^{-1}$, while overall production was 24,946 thousand tons. After rice, wheat is the prime source of calories and the largest protein source [1]. Ninety-four countries and 4.5 billion people in the world are fed by wheat. Wheat provides 21% protein and 20% food calories [2].

Food security is dependent on the production of wheat, which is vital to the diet of people in the Indo-Gangetic plains [3]. This crop is considered the basic food of the

country [4], and about 70% of daily calories are fulfilled by it. Wheat is considered the largest crop in Pakistan because its total cultivated area is almost 40%, and wheat is sown by about 80% of farmers. Worldwide, there is 762 million metric tons of wheat produced yearly [5]. Pakistan ranks as the seventh largest producer of wheat as out of total wheat production, and according to Govt. of Pak., in 2020, the production of wheat in Pakistan was 24,946 thousand tons. There is no doubt that we are making efforts to increase production, but per-acre yield is still lower than expected.

The fact that Pakistan's average production is much lower than that of wheat-producing countries, i.e., Canada, France, the Russian Federation, China, India, and the United States, indicates that improvement is needed [6]. Untimely sowing of wheat, suboptimal seed quality [7], low moisture [8], and improper land preparation [9] are the factors that retard the productivity of wheat. There are two approaches to sowing wheat. One is late sowing of a short-duration crop, while the second is early sowing of a long-duration crop. Wheat grows on 44% of the total cultivated area of Pakistan, out of which 31% is irrigated, and 13% is under rain-fed conditions. Among all cereals, wheat is of primary importance. It feeds about one-third of the worldwide population, delivers half of nutritional protein, and delivers more than half of calories [10].

About a 20–30% loss in profitability and productivity occurs due to expected conventional management practices and cropping systems [11]. Wheat productivity is mainly limited due to late sowing because, after each daily postponement in sowing after the optimal date of sowing, a 36 kg/ha decline in grain yield occurs [12]. Delays in sowing decrease wheat production by up to 2 million tons in various agro-ecological zones of Pakistan [13].

A yield loss of one percent occurs per day as sowing is postponed [14]. If wheat is sown after the third week of November, then yield loss occurs [1]. Late harvesting of previous crops is the main reason for late sowing. When wheat is sown late, inputs applied are not utilized adequately, resulting in low yield [15]. All growth stages of wheat, such as grain filling, tillering, and flowering, are affected when the crop is sown late. When the crop is sown late, the plant faces a high temperature that causes leaf senescence; therefore, the rate of photosynthesis is lowered [16]. This results in a decline in grains per spike and grain weight [12]. Both environmental and internal factors result in the instigation of leaf senescence.

At the anthesis and booting stages, when the temperature is increased up to 37.6 and 37.2 °C, the yield is reduced by up to 50%. Across Central and South Asia and South America, the negative tendency of declining wheat yields and yield losses were observed due to elevated temperature [17]. Elevated temperate tolerance can be incorporated either by breeding or selecting tolerant genotypes or by using agronomic approaches such as the use of plant growth regulators and osmoprotectants [18]. Several growth regulators such as thiourea, betaine, and putrescine are used to surge the yield and growth of crops. Thiourea enhances tolerance against stress because of its high water solubility and quick absorption in living tissues. Thiourea increases vegetative growth, protein contents, and yield in wheat under drought stress [19]. A significant improvement in the growth and photosynthetic efficiency of wheat crop results from the application of thiourea [20,21].

Abiotic stress tolerance of the plants is attributed to different morphological adaptations and molecular mechanisms [22]. Thiourea (TU) has proved itself an efficient osmoprotectant to shield the plants from different abiotic stresses, including drought and heat stress [23]. Tolerance induced by TU is attributed to greater nutrient uptake coupled with the production of osmolytes, improved metabolic processes, and antioxidant defense mechanisms [24].

Endogenous contents of stress-related enzymes are enhanced by salicylic acid [25] which results in higher efficacy of the photosynthetic system and the antioxidant defense mechanism [26], improving stomatal regulation, enhancing root growth, and enhancing water use efficiency [25]. Growth regulators enhance wheat's agronomic performance in various situations. Salicylic acid applied through various methods such as treatment of

seed [27], seed priming [28], and foliar application improves the stress tolerance ability of the crop [29]. For yield-related parameters, the genotypic response of wheat to planting dates differs. Varieties that require more days for heading have a more obvious decline. The potential of a variety at diverse sowing dates depends on its adaptability to different climatic conditions [22].

This study's hypothesis was that thiourea application may enhance the morphological and yield-related attributes, yield, and quality of wheat grains on sandy loam soil of arid and semi-arid regions. Thus, the objective of the present study was to investigate the potential of foliage applied thiourea on the bio-physical, yield, and nutritional-quality-related traits of bread wheat varieties under different growth stages.

## 2. Materials and Methods

### 2.1. Experimental Site and Weather Condition

Two years (2018–2019 and 2019–2020) of field experiments were carried out at the Research Area of Bahauddin Zakariya University, Bahadur Sub Campus Layyah (longitude 70°56′38″ E, latitude 30°57′55″ N, and altitude 143 m from a.s.l). The climatic data of Layyah districts indicate that total rainfalls do not exceed 42 mm per year and mean min. and max. temperatures are 4.9 (in December) and 34.0 °C (in April), respectively (Figure 1). The physio-chemical properties of the experimental soil (characterized as Arenosol—AR) are presented in Table 1.

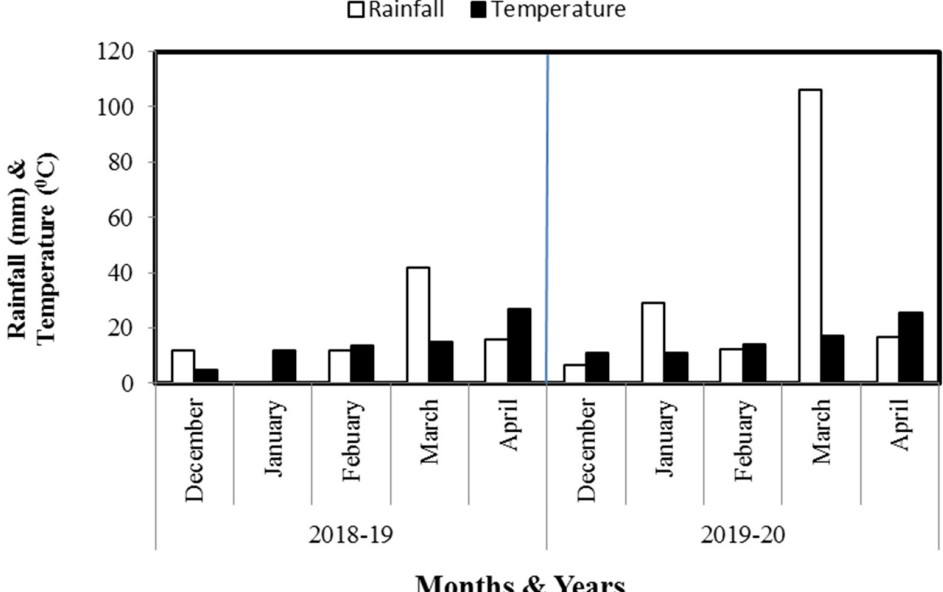

**Figure 1.** Meteorological data of the experimental period (Source: Adaptive Research Centre, Karor Lal Esan, Pakistan).

**Table 1.** Physio-chemical properties of experimental soil (average over 2 years).

| Determinant | Determination Method | Values | Units |
|---|---|---|---|
| Texture of soil | - | Sandy loam | |
| Sand | - | 60 | % |
| Clay | - | 10 | % |
| Silt | - | 30 | % |
| Electrical conductivity | saturation extract method | 1.49 | dSm$^{-1}$ |
| pH | glass electrode making a 1:2.5 soil-water suspension | 8.2 | - |
| Nitrogen | Bremner and Mulvaney, 1982 | 0.4 | % |
| Available phosphorus | Olsen and Watanabe, 1957 | 3 | ppm |
| Available potassium | flame photometry | 96 | ppm |

## 2.2. Experimental Design and Treatments

Wheat seed varieties Galaxy-13 and Gandam-1 were taken from Punjab Seed Corporation, Pvt. Ltd., Punjab, Pakistan. The genotypes were selected based on their best performance in the Punjab region of Pakistan. The chemical (Thiourea) was collected from the Horticulture Laboratory of BZU Bahadur Sub-Campus Layyah, Pakistan. The experiment consisted of three thiourea levels (control, 500, and 1000 mg $L^{-1}$) being applied to two wheat varieties (Galaxy-13 and Gandam-1) at different growth stages (tillering, booting, and heading, i.e., Zodak Stage Z0020, Z0040, and Z0050, respectively). The experimental design was RCBD (with split–split arrangement) replicated three times. The experimental plot size was 8 m × 2 m (16 m$^2$). The crop was sown 25 December 2018–2019 (1st year) and 31 December 2019–2020 (2nd year), respectively, which was approximately one month later than the optimal sowing time (mid to late November).

## 2.3. Crop Husbandry

Two plowings and one planking were performed before sowing the seed. After the preparation of land, pre-sowing irrigation was conducted. The recommended dose of NPK (100:60:40 kg ha$^{-1}$) was applied as a basal dose of whole $P_2O_5$ and $K_2O$, while N was applied in two equal splits, i.e., half at the time of sowing and the other half with the first irrigation. The seed was sown through hand drill when the required moisture level was observed in the soil. At an interval of 15 days, overall, 7 irrigations were completed. Weeds were controlled by manual hoeing. During the complete life cycle of the crop, no attack of disease or insect-pests on the crop was noted. When 80% of spikes became mature in each treatment, the crop was harvested on 16 April 2019 and 10 April 2020.

## 2.4. Observations Recorded

### 2.4.1. Morphological and Yield Related

Ten plants were randomly selected to measure their morphological characteristics (plant height, number of leaves, number of tillers, leaf area (Leaf Area Meter CI-202L, Forestry Suppliers Inc., Jackson, MS, USA), numbers per spike, spikelets per spike, spike length, and yield traits (1000-grain weight, biological yield, and seed yield). An area of 1 m$^2$ was harvested when moisture content reached 20% to measure the biological yield, and then sun-dried for 72 h. After that, the dried samples were used to measure seed yield.

The seed yield of unit area (1 m$^2$) was weighed on an electric balance and later converted into tons per hectare (10,000 m$^2$).

### 2.4.2. Nutritional Quality

Crude protein was determined by the standard micro-Kjeldahl method, by nitrogen determination, and then by multiplying the conversion factor (6.25) [30]. For the determination of grain carbohydrate, standard methods described by [31] Anon (2000) were used, and total fat was determined by following [32] AOAC (2002).

## 2.5. Statistical Analysis

Analysis of variance was performed on the data using Statistics 8.1 software to check the significance of treatment means (thiourea levels, wheat cultivars, and their interactions) for each parameter. The LSD test at 5% probability level was determined by using the above-mentioned software for treatment means [33].

## 3. Results

### 3.1. Morphological Traits

The results reveal that the application of thiourea resulted in significantly higher values of all growth parameters as compared to control. Conversely, wheat varieties showed non-significantly difference during both years. The interactive effect of varieties × thiourea levels × growth stages of wheat was found to be non-significant during the years. Amongst wheat varieties, differences in plant height, number of leaves per plant,

and leaf area remained non-significant during both years of study (Table 2). The maximum number of tillers per plant was recorded in Galaxy-2013 during the 1st year as compared to 2nd year.

**Table 2.** Performance of morphological traits as affected by thiourea and wheat varieties in 2018–2019 and 2019–2020.

| Wheat Varieties | Plant Height (cm) | | Number of Leaves/Plant | | Number of Tillers/Plant | | Leaf Area (cm$^2$) | |
|---|---|---|---|---|---|---|---|---|
| | 2018–2019 | 2019–2020 | 2018–2019 | 2019–2020 | 2018–2019 | 2019–2020 | 2018–2019 | 2019–2020 |
| Gandam-1 | 75.9 | 74.8 | 12.1 | 11.6 | 3.3 B | 8.2 | 18.1 | 17.8 |
| Galaxy-13 | 76.9 | 77.3 | 11.3 | 11.8 | 4.2 A | 8.3 | 18.3 | 18.8 |
| LSD ($p \leq 0.05$) | *ns* | *ns* | *ns* | *ns* | *0.08* | *ns* | *ns* | *ns* |
| Thiourea level (mg L$^{-1}$) | | | | | | | | |
| Control | 71.3 C | 72.3 C | 11.1 | 10.4 C | 4.0 C | 6.3 B | 17.1 C | 16.3 C |
| 500 | 77.2 B | 76.3 B | 11.1 | 11.6 B | 5.8 B | 4.1 C | 18.0 B | 18.0 B |
| 1000 | 80.8 A | 79.6 A | 13.1 | 13.2 A | 6.1 A | 8.1 A | 19.5 A | 20.5 A |
| LSD ($p \leq 0.05$) | *3.35* | *2.93* | *ns* | *0.55* | *0.10* | *0.06* | *0.55* | *0.93* |
| Growth stages | | | | | | | | |
| Tillering | 73.7 B | 72.4 C | 11.3 | 11.3 | 6.7 A | 8.3 A | 17.5 | 18.6 |
| Booting | 78.8 A | 79.2 A | 12.4 | 12.0 | 4.7 B | 5.0 C | 18.1 | 18.4 |
| Heading | 76.5 A | 75.9 B | 11.4 | 11.8 | 4.6 B | 6.1 B | 19.2 | 17.2 |
| LSD ($p \leq 0.05$) | *2.83* | *3.23* | *ns* | *ns* | *0.12* | *0.04* | *ns* | *ns* |

Values sharing same letter in each factor treatments are statistically similar; *ns*—not significant.

Increasing the thiourea levels increased the plant height, the number of leaves/tillers, and leaf area during both years of experimentation (Table 2). Maximum plant height (80.8 and 79.6 cm), number of leaves (13.2 in 2nd year), number of tillers (6.1 and 8.1), and leaf area (19.5 and 20.5 cm$^2$) were observed in plants treated with 1000 mg L$^{-1}$ thiourea during both years in comparison to untreated plants.

Application of thiourea at various growth stages was found to be significant for plant height and number of tillers per plant during both years. Significantly higher plant height was observed at booting stages, while a higher number of tillers was observed at tillering stage during the respective years.

### 3.2. Yield Related Traits

The data pertaining to yield and its attributes as presented in Table 3 reveal that it varied significantly in some regards and non-significantly in others. Application of thiourea significantly affected the yield and yield-related attributes of wheat (Table 3). The interactive effect of varieties × thiourea levels × growth stages of wheat was found to be non-significant over two years. For most of the traits, the minimum value was observed in the control treatment. The difference in the number of spikelets per spike remained non-significant among the genotypes and thiourea treatments (Table 3). The number of grains per spike and 1000-grain weight were recorded at maximum in cv. Galaxy-2013 during the 1st year as compared to 2nd year. The difference in biological yield and grain yield remained non-significant among the wheat varieties during the respective years (Table 3).

Increasing the thiourea levels increased the number of grains per spike in the 1st year and 1000-grain weight, biological yield, and grain yield during both years of experimentation (Table 3). The maximum 1000-grain weight, biological yield, and seed yield were found at the highest rate (1000 mg L$^{-1}$) of thiourea as compared to the control plot during both years.

Application of thiourea at various growth stages was found to be significant for number of grains per spike, biological yield, and grain yield during both years. Significantly, the maximum number of grains, biological yield, and grain yield were observed at heading stage as compared to other growth stages during the respective years.

### 3.3. Nutritional Quality Traits

3.3.1. Total Fats

Application of thiourea at the rate of 1000 mg L$^{-1}$ significantly increased total fats of wheat grain in cv. Gandam-1 as compared to the untreated treatment (control) during both years of the study (Figure 2).

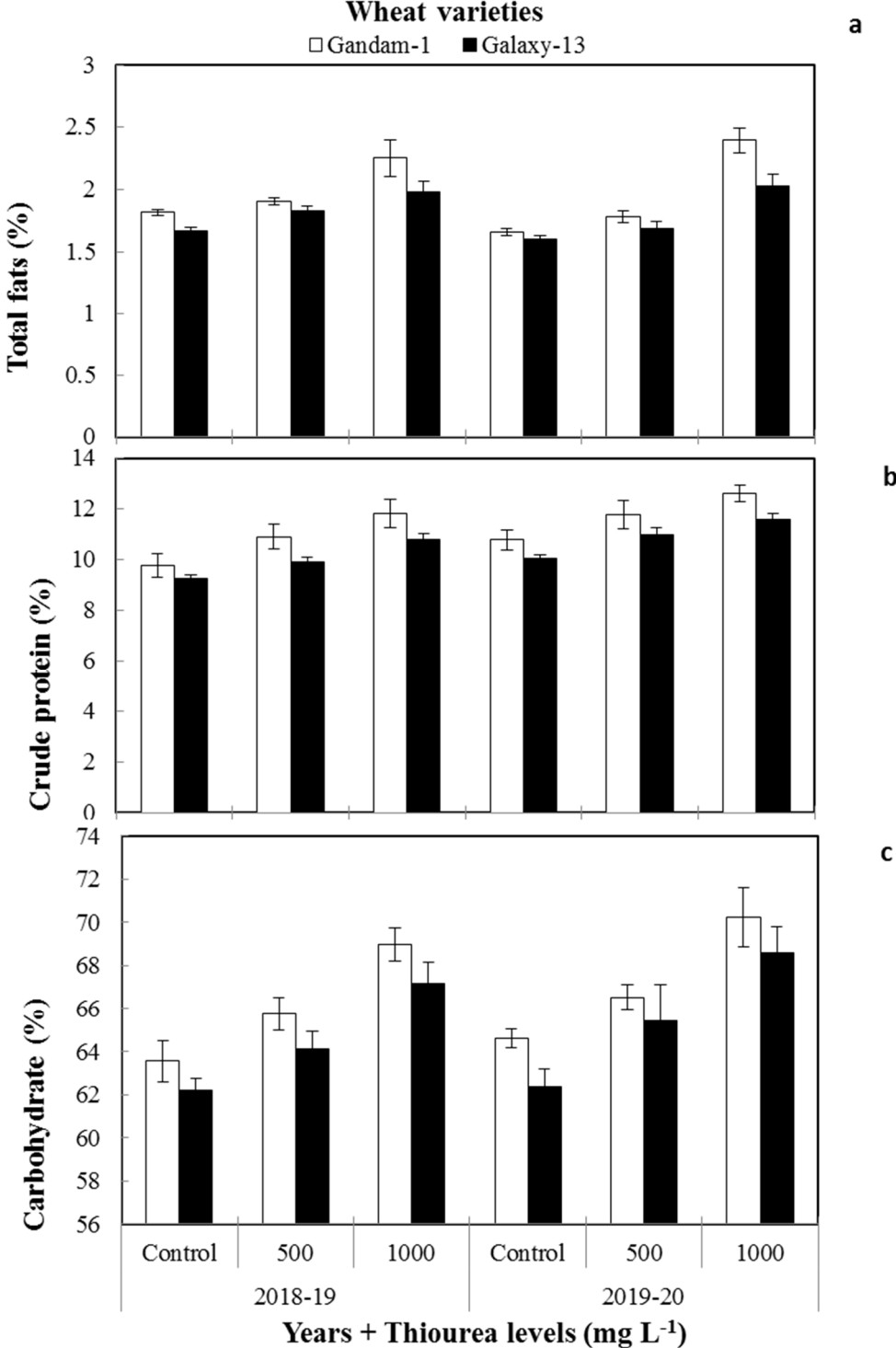

**Figure 2.** Interactive effect of varieties × thiourea on the nutritional quality of bread wheat grains grown in a semi-arid environment. (**a**) total fats; (**b**) crude protein; (**c**) carbohydrate. Error bars represent the standard error (*n* = 5) during 2018–2019 and 2019–2020.

**Table 3.** Performance of different yield-related traits as affected by thiourea and wheat varieties in 2018–2019 and 2019–2020.

| Wheat Varieties | Number of Spikelets/Spike | | Number of Grains/Spike | | 1000-Grain Weight (g) | | Biological Yield (t ha$^{-1}$) | | Seed Yield (t ha$^{-1}$) | |
|---|---|---|---|---|---|---|---|---|---|---|
| | 2018–2019 | 2019–2020 | 2018–2019 | 2019–2020 | 2018–2019 | 2019–2020 | 2018–2019 | 2019–2020 | 2018–2019 | 2019–2020 |
| Gandam-1 | 20.5 | 31.4 | 29.3 B | 30.2 | 33.5 B | 35.0 | 3.42 | 3.02 | 1.95 | 1.99 |
| Galaxy-13 | 20.6 | 27.8 | 31.5 A | 30.6 | 34.7 A | 35.4 | 3.61 | 3.37 | 2.14 | 2.10 |
| LSD ($p \leq 0.05$) | *ns* | *ns* | *1.86* | *ns* | *0.68* | *ns* | *ns* | *ns* | *ns* | *ns* |
| Thiourea level (mg L$^{-1}$) | | | | | | | | | | |
| Control | 19.9 | 19.0 | 27.9 B | 29.9 | 33.8 B | 33.7 B | 3.70 C | 3.36 C | 1.69 C | 2.07 C |
| 500 | 20.2 | 32.3 | 31.0 A | 30.4 | 33.7 B | 34.8 B | 3.86 B | 3.93 B | 2.07 B | 2.18 B |
| 1000 | 21.5 | 37.5 | 33.2 A | 31.0 | 34.7 A | 37.2 A | 3.98 A | 4.10 A | 2.37 A | 2.32 A |
| LSD ($p \leq 0.05$) | *ns* | *ns* | *1.46* | *ns* | *0.83* | *1.47* | *0.09* | *0.15* | *0.20* | *0.10* |
| Growth stages | | | | | | | | | | |
| Tillering | 21 | 28.5 | 29.6 B | 27.2 C | 34.0 | 35.5 | 3.34 B | 3.39 C | 1.91 B | 1.97 C |
| Booting | 20.5 | 21.8 | 30.4 AB | 31.9 B | 34.2 | 34.8 | 3.67 B | 3.59 B | 2.02 B | 2.11 B |
| Heading | 20.1 | 38.0 | 31.2 A | 33.2 A | 34.1 | 35.3 | 3.95 A | 3.72 A | 2.17 A | 2.23 A |
| LSD ($p \leq 0.05$) | *ns* | *ns* | *0.96* | *1.20* | *ns* | *ns* | *0.25* | *0.12* | *0.10* | *0.11* |

Values sharing same letter in each factor treatments are statistically similar; *ns*—not significant.

### 3.3.2. Crude Protein

The interaction between different levels of the thiourea and wheat cultivars indicates that the highest crude protein contents were obtained wheat grain in cv. Gandam-1 at 1000 mg L$^{-1}$ thiourea level as compared to the other treatments during both years (Figure 2).

### 3.3.3. Carbohydrate

Data presented in Figure 2 show that exogenous application of thiourea at 1000 mg L$^{-1}$ concentration significantly increased the total carbohydrate in wheat grain cv. Gandam-1 as compared to other treatments during both years (Figure 2).

## 4. Discussion

Thiourea is an important plant osmoprotectant used for sulfur fertilization and abiotic stress tolerance [34]. The role of TU in the cross-talk of plants for hormonal signaling and regulation of gene expression in abiotic stress has also been reported. Thiourea improves the yield of stressed plants by regulating the gene expression of physiological and antioxidant defense mechanisms [35–37]. The role of TU has also been reported in the regulation of many morpho-physiological processes such as carbohydrate metabolism, gene regulation, breaking of seed dormancy, improving seed germination, root–shoot growth, and yield components [38–42].

Improvement in crop allometric traits such as leaf area index and morphological traits with thiourea application might be due to better translocation of photosynthates and an increase in the ability of plants to withstand abiotic stress, particularly heat stress that is common at the maturity stage of wheat crop [43–45]. These results are strengthened by previous findings of Amin et al. [22] who also reported that foliar feeding of thiourea significantly increased the growth-related parameters of faba bean. The application of thiourea ameliorated heat stress and improved photosynthetic activities, resulting in vigorous growth and enhancement in plant height [38,46]. Khanna et al. [35] also reported that thiourea application substantially improved the number of spikelets and grains per spike by increasing enzymatic (rubisco) and photosynthetic activities and plants' resistance against abiotic stresses, particularly heat stress. These results are supported by previous findings of Sanaullah et al. [47], who reported a significant increase in grain weight with thiourea application, which might be due to better photosynthesis and translocation of starch towards the developing kernel. Improvement in biological and grain yield with foliar-applied thiourea in this study was consistent with the earlier reports [35]. Similarly, a remarkable increase in harvest index was also seen with the application of thiourea. This increase in harvest index might be due to the combined increase in grain and biological

yield of wheat. These results are supported by previous findings of Asthir et al. [40], who reported a substantial increase in harvest index with the application of thiourea.

The study indicated that application of thiourea at different levels at various growth stages improves the wheat cultivars' growth rates, morphological traits (plant height, number of tillers, number of leaves per plant, leaf area, number of spikelets, and number of grains per spike), and yield-related traits (thousand-grain weight, biological yield, and grain yield) (Table 3). It has been observed that wheat cultivar Galaxy-2013 had the maximum plant height, leaf area, number of grains per spike, and biological yield during both years and also had the maximum thousand grain weight and grain yield during the second year (Tables 2 and 3).

The application of thiourea at the rate of 1000 mg $L^{-1}$ resulted in the maximum plant height, number of leaves per plant, number of spikelets per spike, number of fertile tillers, and yield-related traits during both years, and also resulted in the maximum number of grains per spike and leaf area during the 1st and 2nd year, respectively (Tables 2 and 3). The improved yield and related traits with the application of thiourea is attributed to increased availability of N in TU, which improves the photosynthetic efficiency and ultimately yield [48]. Moreover, sulfur is also an important component of plant nutrition, which is often ignored. Availability of sulfur also increased with TU application, leading to higher yield. Moreover, availability of sulfur improves the N metabolism and expression of stress-related genes that regulate growth during stress [37,38,49] faced by wheat due to late sowing.

At the growth stage tillering, the morphological traits (plant height, number of tillers, leaf area, and number of grains per spike) and yield-related traits (thousand-grain weight and grain yield) were recorded at maximum during the second year [50]. According to Thapa et al. [51], when the crop is sown late, due to reduction in yield-related characteristics such as the number of grains per spike, grain yield, and number of tillers, overall yield is reduced. According to Marasini et al. [52], in comparison to late sowing, normal sowing gave higher yield. At 39 kg $ha^{-1}$ per day, everyday reduction occurs in grain yield as sowing is postponed from 20th November [53]. Oakes et al. [54] observed that as compared to late sowing, early sowing improved spikelets per spike, grain weight and germination per unit area, plant height, and grains per spike. In the past, various high-yielding cultivars were made, but unfortunately, due to variations in several environmental and edaphic situations, these have lost their potential. Therefore, present moment demands that we make varieties that are high yielding as well as more adaptable to environmental situations. This will help to increase crop yield.

Varietal selection and sowing time are the most important factors determining the yield of wheat. For flowering, emergence, and growth, wheat requires a certain amount of light and temperature [22,55]. If wheat is sown before the optimum sowing date, then the plants produced are weak. There are many harmful effects of high temperature. Decaying of endosperm due to activities of fungi or bacteria, irregular germination, and the death of embryos are some harsh effects of high temperature. Poor tillering and slow growth of crop result from late sowing, although if a variety has a short duration at the grain filling stage, the plant can escape from high temperatures [56].

Improvement in the nutritional quality of grains due to thiourea is attributed to improvement in the mineral nutrition by application of thiourea [57]. It is reported that TU regulates the *n* and *p* metabolism and improves the photosynthesis and ultimately concentration of starch, soluble proteins, and fats [58], which strengthens our findings.

## 5. Conclusions

In conclusion, exogenous application of thiourea improves the morphological attributes, yield, and yield-related attributes of wheat in semi-arid regions. In addition, thiourea also enhances the nutritional quality of wheat grains. Therefore, thiourea application is found to be the best strategy to improve the productivity of wheat grown on sandy loam soils in semi-arid regions; however, for wider recommendations, more trials may be conducted across various agro-ecological regions.

**Author Contributions:** A.S. (Ahmad Sher) and A.S. (Abdul Sattar) conceived of the idea. A.S. (Ahmad Sher) and M.N. conducted the experiment and collected the literature review. S.U.-A. and M.I. provided technical expertise to strengthen the basic idea. Y.B., A.M., X.W., and A.Q. helped in statistical analysis. S.F., A.S. (Abdul Sattar), and A.Q. proofread and provided intellectual guidance. All authors read the first draft, helped in revision, and approved the article. All authors have read and agreed to the published version of the manuscript.

**Funding:** The publication of the present work is supported by the Natural Science Basic Research Program of Shaanxi Province (grant No. 2018JQ5218) and the National Natural Science Foundation of China (51809224), Top Young Talents of Shaanxi Special Support Program.

**Institutional Review Board Statement:** Not applicable.

**Informed Consent Statement:** Not applicable.

**Data Availability Statement:** All relevant data is available within the manuscript in the form of tables and figures.

**Conflicts of Interest:** The authors declare that there are no conflicts of interest.

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
