# Peer review of "Exogenous Application of Thiourea for Improving the Productivity and Nutritional Quality of Bread Wheat (Triticum aestivum L.)"

_agronomy, doi:10.3390/agronomy11071432_

Round 1
Reviewer 1 Report
The authors addressed all of my comments to a satisfactory degree. I have no further feedback to this manuscript. In my opinion it is suitable for publication in its current form.
Reviewer 2 Report
Dear Authors
I would like to thank the Authors of the manuscript (Exogenous Applied Thiourea for Improving the Productivity and Nutritional Quality of Bread Wheat (Triticum aestivum L.) for referring to my comments and making corrections to the manuscript, which I accept.
Kind regards,
Reviewer 3 Report
The comments from the first review have been corrected.The work is suitable for publication in Agronomy
This manuscript is a resubmission of an earlier submission. The following is a list of the peer review reports and author responses from that submission.
Round 1
Reviewer 1 Report
I find this paper interesting and valuable as it provides new scientific evidence , However, I have a few remarks that should be considered by the authors
- The title should be changed, for example:
Exogenously Applied Thiourea for Improving the Productivity and Nutritional Quality of Bread Wheat (Triticum aestivum L.)
or
Exogenous application of Thiourea for Improving the Productivity and Nutritional Quality of Bread Wheat (Triticum aestivum L.)
- This sentence is included in both: the abstract and the conclusions : “however for the recommendation of wheat cultivars, specific area and appropriate sowing time may also be considered”. This sentence is not clear and should be rephrased.
- In the material and methods chapter, the characteristics of the varieties should be presented. Please provide reason explaining why these varieties were chosen for the experiment
- It is not clear why the authors refer to delayed sowing term throughout the manuscript even though the sowing term was not one of the experimental factors. Is it that sowing on December 25th and December 31st 2018 and 2019 are considered as delayed sowing terms? If that’s the case, I recommend explaining this in the method section by emphasizing that these are delayed sowing terms. It is then also necessary state what is the optimal sowing term.
- The Authors write that the maximum numbers of spikelets were recorded in wheat cultivar “Gandam-1” which was statistically at par with “Glaxay-2013” during the first year. Please correct this sentence.
- in the description of the results concerning the effect of the dose of thiourea on the yield and yield characteristics (tab.3) there is no reference to the control. Please add the information regarding control. Additionally, the text lacks a description of the effect of using the preparation in different phases on the yield and yield structure features.
- In the text, the authors incorrectly refer to table 4 as the reference should be to table 3.
Author Response
Dated: July 4, 2021
Dear Editor,
Greetings,
Thank you very much for your time and comments regarding our manuscript (agronomy-1245266). Our manuscript “Exogenous application of Thiourea for Improving the Productivity and Nutritional Quality of Bread Wheat (Triticum aestivum L.)” has been revised carefully and here we are giving our response to the reviewers’ comments. We have improved the manuscript according to the reviewers’ comments and suggestions. All the revisions can be easily identified from manuscript highlighted with yellow color.
Once again thanks for your co-operation and valuable comments and suggestion. Moreover, the efforts of the reviewer are highly appreciated. We are hoping for pleasant response and further good comments (if any) from your side.
Dr. Abdul Qayyum
Department of Agronomy,
The University of Haripur 22620 Pakistan
*********************************************************************
We are thankful to editor and reviewers for timely completion of review process and providing us with valuable feedback.
Reviewer 1(Red color in manuscript file):
- Comment: The title should be changed, for example?
Response: Title has been revised as per suggestions
- This sentence is included in both: the abstract and the conclusions: “however for the recommendation of wheat cultivars, specific area and appropriate sowing time may also be considered”. This sentence is not clear and should be rephrased.
Response: The sentences has been revised for the clarity (Line 25-27 and conclusion section)
- Comment: In the material and methods chapter, the characteristics of the varieties should be presented. Please provide reason explaining why these varieties were chosen for the experiment
Response: Relevant information has been added (Line 119-120)
- Comment: It is not clear why the authors refer to delayed sowing term throughout the manuscript even though the sowing term was not one of the experimental factors. Is it that sowing on December 25th and December 31st 2018 and 2019 are considered as delayed sowing terms? If that’s the case, I recommend explaining this in the method section by emphasizing that these are delayed sowing terms. It is then also necessary state what is the optimal sowing term.
Response: Relevant information has been added (Line 126-128)
- Comment: The Authors write that the maximum numbers of spikelets were recorded in wheat cultivar “Gandam-1” which was statistically at par with “Glaxay-2013” during the first year. Please correct this sentence.
Response: Results section have been revised thoroughly as per instructions (Line 185-187)
- Comment: in the description of the results concerning the effect of the dose of thiourea on the yield and yield characteristics (tab.3) there is no reference to the control. Please add the information regarding control. Additionally, the text lacks a description of the effect of using the preparation in different phases on the yield and yield structure features.
Response: Relevant information has been added in relevant section
- Comment: In the text, the authors incorrectly refer to table 4 as the reference should be to table 3.
Response: Relevant information has been corrected.

Reviewer 2 Report
Dear authors,
Thank you for the opportunity to review your manuscript, but this manuscript is not innovative enough.
The authors assessed the effect of Exogenous Applied Thiourea for Improving the Productivity and Nutritional Quality of Bread Wheat (Triticum aestivum L.). Previous studies arleady used these methods in majority.
Below I present comments to the manuscript:
Abstract:
In the Abstract section ,authors only described the content of the study, not the significance of the study.
Material and methods:
- Why were these cultivars selected for research? Whether they are popular for cultivation in the research area?
- Please provide the scale of development phases (scale BBCH or Zadoks?? Or other?),
- Please provide the soil type it according to ......WRB, Or other? (WRB 2014 or 2015. World Reference Base for Soil Resources 2014. International soil classification system, for naming soils and creating legends for soil maps. World Soil Resources Reports, 106. )
- Table 1., Please provide methods physio-chemical properties of experimental soil,
- How was nitrogen (and PK) fertilization dosed? in how many doses? At what stage of growth of plant?
- Chapter 2.4.1. Morphological and yield related, how the leaf area measurement was performed? what device was used for the measurement? What was the grain moisture at harvest?
Results, Discussion:
- Please explain in the Results or Discussion what factors (abiotic or biotic) caused the low grain yield (from 1.60 to 4.42 t.ha-1) of the tested wheat cultivars,
- The Table 2 i 3 in the manuscript are not clear enough, this applies to the same LSD values for the experimental factors, it is unclear. In addition, there is no interaction for experience factors.
- Table 2, In biological yield and Seed Yield - whether LSD is calculated correctly, LSD values is too high. Has the statistical program been selected correctly?
- Results should write again after the improvement of the statistical calculations.
Conclusions: In the Conclusions section, the authors also did not elaborate on the significance of the study.
References: Please, prepare a literature record according to the requirements of the Agronomy
Author Response
Dated: July 4, 2021
Dear Editor,
Greetings,
Thank you very much for your time and comments regarding our manuscript (agronomy-1245266). Our manuscript “Exogenous application of Thiourea for Improving the Productivity and Nutritional Quality of Bread Wheat (Triticum aestivum L.)” has been revised carefully and here we are giving our response to the reviewers’ comments. We have improved the manuscript according to the reviewers’ comments and suggestions. All the revisions can be easily identified from manuscript highlighted with yellow color.
Once again thanks for your co-operation and valuable comments and suggestion. Moreover, the efforts of the reviewer are highly appreciated. We are hoping for pleasant response and further good comments (if any) from your side.
Dr. Abdul Qayyum
Department of Agronomy,
The University of Haripur 22620 Pakistan
*********************************************************************
We are thankful to editor and reviewers for timely completion of review process and providing us with valuable feedback.
Reviewer 2 (Blue color in manuscript file):
- Comment: Why were these cultivars selected for research? Whether they are popular for cultivation in the research area?
Response: explanation has been added in materials and method at relevant section (Line 119-120)
- Please provide the scale of development phases (scale BBCH or Zadoks?? Or other?),
Response: explanation has been added in materials and method at relevant section (Line 123-126)
- Comment: Please provide the soil type it according to ......WRB, Or other? (WRB 2014 or 2015. World Reference Base for Soil Resources 2014. International soil classification system, for naming soils and creating legends for soil maps. World Soil Resources Reports, 106. )
Response: Information has been added in materials and method at relevant section (Line 111)
- Comment: Table 1., Please provide methods physio-chemical properties of experimental soil,
Response: Information has been added in materials and method at relevant section (table 1)
- Comment: How was nitrogen (and PK) fertilization dosed? in how many doses? At what stage of growth of plant?
Response: Information has been added in materials and method at relevant section (Line 131-134)
- Chapter 2.4.1. Morphological and yield related, how the leaf area measurement was performed? what device was used for the measurement? What was the grain moisture at harvest?
Response: Information has been added in materials and method at relevant section (line 142-145)
Results, Discussion:
- Comment: Please explain in the Results or Discussion what factors (abiotic or biotic) caused the low grain yield (from 1.60 to 4.42 t.ha-1) of the tested wheat cultivars,
Response: Information has been added in results section and also discussed in discussion section
- Comment: The Table 2 i 3 in the manuscript are not clear enough, this applies to the same LSD values for the experimental factors, it is unclear. In addition, there is no interaction for experience factors.
Response: Information has been added in table 2, 3 at relevant section (table 2, 3)
- Comment: Table 2, In biological yield and Seed Yield - whether LSD is calculated correctly, LSD values is too high. Has the statistical program been selected correctly?
Response: This was a typo error, has been corrected now
- Comment: Results should write again after the improvement of the statistical calculations.
Response: whole results have been improved
- Conclusions: In the Conclusions section, the authors also did not elaborate on the significance of the study.
Response: conclusion section has been revised thoroughly
References: Please, prepare a literature record according to the requirements of the Agronomy
Response: References has been updated as per journal formatting

Reviewer 3 Report
The paper describes research on the influence of thiourea applied as foliar on the yielding characteristics and quality of wheat grown in semi-arid areas.
The layout of the work is correct. Divided into introduction, research methods, results, discussion and conclusions. At the end of the work, there is a references. The test results are presented in 2 tables and 1 figure. The abstract is correctly written. It contains a summary of the conducted research and results..
However, numerous errors were found in the work that should be corrected and re-reviewed.
- There is no research hypothesis in the introduction, it should be added.
- Materials and methods should be corrected. Add methods of assessing soil conditions. In the subsection (crop husbandry), the date of sowing wheat in 2019-2020 is missing. It should be added how the soil moisture was assessed. In the chapter (morphological and yield related), the grain moisture and how it was converted into the yield of 1ha should be given. Write down methods for the determination of fat, protein and carbohydrates. In the stastical analysis section, describe how exactly the LSD α = 0.05 was calculated
- The distribution of research results needs to be improved. There are errors in tables 2 and 3. In table 2 (no homogeneous groups (in number of levels / plant in 2018/2019 and 2019/2020 years). Doubts are raised by the large number of leaves and very high tillers (in the literature, wheat tillers is on average 4-6) and the table is 13-20 (results need to be checked) Why are the same LSD values for Thiourea level and Growth stages levels ? The table 3 needs to be revised. There are no homogeneous groups in Table 3 to compare the results over the years of the study. The errors are in the LSD (seed yield) values - 6.20 and 11.33 it is impossible with the yield of 1.91 t ha-1 and 2.20 t ha-1 in 2018/2019 and 2.66 t ha-1 and 3.32 t ha-1 in 2019/2020. They are also in the lower line of LSD (Thiourea level) on the biological yield and seed yield and (Growth stages). Doubts raise equal LSD values. The results after the improvement should be described in more detail.
- The discussion of the results largely does not refer to the research results. It needs to be corrected.
- Conclusions should be improved. They do not relate to the purpose of the research.
- The 24 references were not cited in the manuscript text and 6 references were repeated twice.
- Correction is required in English by a professional translator.
The work is an original research study, but contains many errors and must be reviewed again before publication in Agronomy. It is not suitable for publication in this version

Author Response
Dated: July 4, 2021
Dear Editor,
Greetings,
Thank you very much for your time and comments regarding our manuscript (agronomy-1245266). Our manuscript “Exogenous application of Thiourea for Improving the Productivity and Nutritional Quality of Bread Wheat (Triticum aestivum L.)” has been revised carefully and here we are giving our response to the reviewers’ comments. We have improved the manuscript according to the reviewers’ comments and suggestions. All the revisions can be easily identified from manuscript highlighted with yellow color.
Once again thanks for your co-operation and valuable comments and suggestion. Moreover, the efforts of the reviewer are highly appreciated. We are hoping for pleasant response and further good comments (if any) from your side.
Dr. Abdul Qayyum
Department of Agronomy,
The University of Haripur 22620 Pakistan
*********************************************************************
We are thankful to editor and reviewers for timely completion of review process and providing us with valuable feedback.
Reviewer 3 (Green color in manuscript file)::
- Comment: There is no research hypothesis in the introduction, it should be added.
Response: Information has been added in relevant section (line 99-103)
- Materials and methods should be corrected. Add methods of assessing soil conditions. In the subsection (crop husbandry), the date of sowing wheat in 2019-2020 is missing. It should be added how the soil moisture was assessed. In the chapter (morphological and yield related), the grain moisture and how it was converted into the yield of 1ha should be given. Write down methods for the determination of fat, protein and carbohydrates. In the statistical analysis section, describe how exactly the LSD α = 0.05 was calculated
Response: Information has been added in relevant section of materials and methods (Line 125-127 and 146-159)
- Comment: The distribution of research results needs to be improved. There are errors in tables 2 and 3. In table 2 (no homogeneous groups (in number of levels / plant in 2018/2019 and 2019/2020 years). Doubts are raised by the large number of leaves and very high tillers (in the literature, wheat tillers is on average 4-6) and the table is 13-20 (results need to be checked) Why are the same LSD values for Thiourea level and Growth stages levels ? The table 3 needs to be revised. There are no homogeneous groups in Table 3 to compare the results over the years of the study. The errors are in the LSD (seed yield) values - 6.20 and 11.33 it is impossible with the yield of 1.91 t ha-1 and 2.20 t ha-1 in 2018/2019 and 2.66 t ha-1 and 3.32 t ha-1 in 2019/2020. They are also in the lower line of LSD (Thiourea level) on the biological yield and seed yield and (Growth stages). Doubts raise equal LSD values. The results after the improvement should be described in more detail.
Response: All the corrections have been incorporated in relevant sections and highlighted with green color
- Comment: The discussion of the results largely does not refer to the research results. It needs to be corrected.
Response: Whole results and discussion section have revised
- Comment: Conclusions should be improved. They do not relate to the purpose of the research.
Response: Whole conclusion section have revised
- Comment: The 24 references were not cited in the manuscript text and 6 references were repeated twice.
Response: All the missing reference have been added and cross checked
- Comment: Correction is required in English by a professional translator.
Response: English language have been improved by a native speaker
